# Physical and Physiological Predictors of FRAN CrossFit^®^ WOD Athlete’s Performance

**DOI:** 10.3390/ijerph18084070

**Published:** 2021-04-12

**Authors:** Luis Leitão, Marcelo Dias, Yuri Campos, João Guilherme Vieira, Leandro Sant’Ana, Luiz Guilherme Telles, Carlos Tavares, Mauro Mazini, Jefferson Novaes, Jeferson Vianna

**Affiliations:** 1Sciences and Technology Department, Superior School of Education of Polytechnic Institute of Setubal, 2910-761 Setúbal, Portugal; 2Life Quality Research Centre, 2040-413 Rio Maior, Portugal; 3Post Graduate Program in Physical Education, Federal University of Juiz de Fora, São Pedro 36036-900, Brazil; diasmr@gmail.com (M.D.); reiclauy@hotmail.com (Y.C.); joaoguilhermevds@gmail.com (J.G.V.); leandrosantana.edufisica@hotmail.com (L.S.); jeferson.vianna@ufjf.edu.br (J.V.); 4Laboratory of Exercise Physiology and Morphofunctional Assessment, Granbery Methodist College, Juiz de Fora 36010-530, Brazil; 5Study Group and Research in Neuromuscular Responses, Federal University of Lavras, Aquenta Sol, Lavras 37200-900, Brazil; 6School of Sports and Physical Education, Federal University of Rio de Janeiro, Rio de Janeiro 21941-901, Brazil; guilhermetellesfoa@hotmail.com (L.G.T.); jeffsnovaes@gmail.com (J.N.); 7Graduate Program of Physical Education of Sudamerica Faculty, Cataguases 36774-552, Brazil; carlos.gontav@live.com (C.T.); personalmau@hotmail.com (M.M.)

**Keywords:** high-intensity functional training, strength endurance, maximal strength, blood lactate, VO_2max_, RPE

## Abstract

CrossFit^®^ training is one of the fastest-growing fitness activities in the world due to its varied functional movement and competition experience. The performance is present in almost every workout of the day (WOD); however, there is a lack of knowledge in the science that did not allow us to fully understand the performance determinants of CrossFit WOD’s like we do for other individual or team sports. The purpose of this study was to analyze the physical and physiological variables of recreational trained CrossFit athletes during one of the most famous WOD, FRAN, and to identify which variables best determine performance. Methods: Fifteen CrossFit practitioners performed, alone on separate days, 1RM and a maximum of repetitions of pull-ups test, 1RM and a maximum of repetitions of thrusters with 95 lb/43.2 kg, FRAN CrossFit WOD, and 2K Row test. Results: Blood lactate concentrate, HR_max_, HR_av_, and RPE achieved higher values for 2K Row and maximum repetitions of thrusters. Maximum repetition of thrusters and pull-ups, 1RM of thrusters, and 2K Row resulted in moderate to strong correlation with FRAN performance (*r* = −0.78; *r* = −0.58; *r* = −0.67; *r* = 0.63, respectively). Conclusions and practical applications: FRAN performance was strongly related to maximal and endurance strength training of thrusters, which should be prioritized.

## 1. Introduction

CrossFit^®^ training is an alternative modality to high-intensity functional training (HIFT). Due to its constantly varied functional movements performed at relatively high intensity through metabolic conditioning, gymnastics, and weightlifting [1], this modality has been increasing in its popularity across the world.

The basic tasks are consisted of little to no resting periods during the activity in order to complete a task as fast as possible (for time) or achieve the greatest number of repetitions in a certain period of time (as many repetitions as possible, AMRAP). The high intensity of CrossFit workouts improves muscular endurance and strength, fitness level, as well as body composition [2,3,4,5].

The huge number of participants and increasing competition and professional athletes of CrossFit allowed us to characterize it as a training program and sport of fitness. In this case, the role of science is to clarify doubts about the key performance indicators [4,5,6,7].

However, few studies investigated performance predictors of CrossFit. Butcher et al. [8] verified that GRACE and FRAN (two types of workouts) strongly correlated with the strength data (CrossFit Total benchmark) and oxygen consumption at the anaerobic threshold. Bellar et al. [9] showed that a higher aerobic capacity and peak power and younger ages were associated with a higher numbers of repetitions. Recently, Crawford et al. [10] showed that there were significant associations between predictor variables (VO_2max_, 1RM, and power) and work capacity, and Landero-Gómez and Menacho-Juan [6] showed that strength, muscle mass, low adiposity, and aerobic capacity were important components that characterized CrossFit competitors. FRAN workout of the day (WOD) is one of the most famous type of training that every CrossFit athlete does to control their performance improvements [6,11]. According to Zeitz et al. [11], 33% of the variance of performance in a modified FRAN can be explained by the total strength of CrossFit Total.

Therefore, the purpose of this study was (1) to analyze the physical and physiological variables of recreational trained CrossFit athletes during FRAN WOD and (2) to identify which variables best determine performance. We hypothesized that the neuromuscular system (maximal strength and strength endurance) would be the predictor of performance in FRAN CrossFit WOD.

## 2. Materials and Methods

### 2.1. Subjects

Fifteen male CrossFit amateur athletes (with minimum three years of experience/four sessions a week, 24.03 ± 4.2 years, 78.2 ± 10.59 kg, 1.75 ± 0.07 m, 25.82 ± 2.7 kg/m^2^, 19.39 ± 4.8 body fat percentage) free of injury and known illness volunteered in this study. All the subjects trained in the same CrossFit affiliate gym with the same training periodization at least over the last three years and were advised to sleep between six and eight hours the night before each experimental session; to maintain their regular hydration and food consumption habits; to avoid any exercise in the 48 h before the experimental sessions; and to avoid smoking, alcohol, and caffeine consumption for 24 h before the experimental session. Before all the assessments, the participants attended a descriptive session about the exercises and tests that they would perform. All volunteers signed a written informed consent document. The experimental protocol was approved by the local ethics committee (3749878/2019) and was performed according to the Declaration of Helsinki.

### 2.2. Procedures

Figure 1 illustrates the organization of the study, in which each volunteer underwent four days of assessments under the same environmental conditions (temperature 22–24 °C, humidity 55–65%, 16:00–18:00 h) and was supervised by the same experienced investigator (ensuring the correct completion of each movement). All the following measurements were conducted at a full equipped CrossFit Box (where the volunteers trained) with a 48 h interval between each measurement: (1) 1RM and maximum repetitions of pull-ups test, (2) 1RM and maximum repetitions of thrusters with 95 lb/43.2 kg, (3) FRAN CrossFit WOD, and (4) 2K Row test.

#### 2.2.1. FRAN

This CrossFit benchmark WOD was a combination of barbell thrusters (a front squat followed by a push press with 43.2 kg), and pull-ups performed in a 21-15-9 repetition scheme, where the athlete performed 21 thrusters and 21 pull-ups, then 15 thrusters and 15 pull-ups, then 9 thrusters and 9 pull-ups as fast as possible [12]. Variations of pull-ups, including butterfly and kipping, were valid. The assessment started with a warm-up of 5 min of joint mobility and dynamic stretching exercises and a specific warm-up of 10 repetitions with a self-low load of each movement of FRAN. After 5 min of rest, the FRAN protocol was performed. Blood lactate, rating of perceived exertion (RPE), maximum heart rate (HR_max_), average heart rate (HR_av_), and time performed were recorded.

#### 2.2.2. RM Pull-Up and Thruster Measurements

Maximal strength of pull-up and thrusters was assessed through the 1RM test using a 300 g belt and incremental plates (Eleiko, Halmstad, Sweden) for the pull-up and a 20 kg barbell and incremental plates (Eleiko, Halmstad, Sweden) for the thrusters.

The pull-up was performed starting from the bar with the elbows fully extended and hands in pronation separated by a distance wider than the hips and with a belt holding the load. The movement started by bending the elbows and raising the shoulders until the chin was higher than the bar. No pull-ups variations, including butterfly and kipping, were valid, only strict pull-ups.

The warm-up consisted of 5 min of joint mobility and dynamic stretching exercises and a specific warm-up of two sets of 8 repetitions with body-weight for pull-up and a self-selected low load for the thrusters, both followed by 5 min of rest. Then, the initial load was estimated based on the participants’ training history. If the athlete failed to perform 1 repetition or performed more than 1, the load was adjusted by a minimum of 1 kg. The rest in between attempts was established at 5 min, and no participant needed more than three attempts to reach 1RM. The RPE and the weight lifted were recorded.

#### 2.2.3. Maximum Number of Pull-Ups and Thrusters Measurements

The maximum number of non-stop pull-ups and thrusters were carried out until concentric failure. For pull-ups, kipping and butterfly pull-up movements were allowed, and only the pull-ups performed with the chin over the bar were valid. For thrusters, the bar started from the floor (performed with the specific FRAN load of 43.2 kg/95 lb) and only the repetitions of the front squat during which the hips reached below parallel at the bottom body followed by a full extension of the entire body after the push press were valid. The number of repetitions, blood lactate, RPE, HR_max_, and HR_av_ were recorded.

#### 2.2.4. 2K Row Test

The 2K Row test consisted of a time-trial of 2000 m rowing, after a warm-up of 5 min, followed by a 5 min rest interval. Maximum aerobic capacity was estimated according to the work average (W_av_) attained in the performance of 2000 m row test on a Concept 2 ergometer [13,14]:VO_2max_ (l/min) = 1.631 + 0.0088 W_av_

Work average, blood lactate, RPE, HR_max_, HR_av_, and time performed was recorded.

#### 2.2.5. Anthropometric Evaluation

Height and body mass were measured with a scale OMRON BF 303 (OMRON Healthcare Europe BV, Matsusaka, Japan), with a stadiometer (Seca, Hamburg, Germany) and bioelectrical impedance analysis to assess the weight (kg), height (cm), and body fat percentage (%BF, %).

#### 2.2.6. Blood Lactate Assessment

Capillary blood samples were collected after the first drop of blood was dismissed through a transcutaneous puncture on the medial side of the tip of the middle finger using a disposable hypodermic lancet (Accu-Chek Safe-T-Pro Uno, Roche^®^, Hawthorne, CA, USA). The blood lactate concentration was measured by photometric reflectance on a validated portable lactate analyzer (Accusport, Boehringer Mannheim—Roche^®^, Hawthorne, CA, USA). Before the tests, the lactate analyzer was calibrated with different standard solutions of known lactate concentrations (2, 4, 8, and 10 mmol L^–1^). Blood lactate concentrations were measured two minutes after FRAN, 2K Row, maximum repetitions test of thrusters and pull-ups.

#### 2.2.7. Heart Rate Monitoring

Continuous monitoring of the heart rate (HR) during every test was done with the use of a Garmin HRM-Run strap and monitor (Garmin Fenix 3). The maximum and average heart rate (HR_max_ and HR_av_) was registered from FRAN, maximum repetition of pull-ups and thrusters test, and 2K Row test. HR_max_ was the highest heart rate value attained by the athlete, and HR_av_ the heart rate mean from the beginning to the end of each test using Garmin connect software.

#### 2.2.8. Rating of Perceived Exertion (RPE)

RPE was obtained using the CR10 Borg RPE scale [15,16]. The scale was explained before the exercise and was recorded 2 min after the end of each test.

### 2.3. Statistical Analysis

Descriptive procedures of central tendency and dispersion were used to characterize the variable values, and the Shapiro–Wilk test was applied to verify the normal distribution of the data. Spearman’s correlation was used to analyze all the correlations between study variables. The significance level was *p* ≤ 0.05, and the software used for data analysis was SPSS 26.0 for Windows (SPSS Inc., Chicago, IL, USA).

## 3. Results

The results of FRAN and the physiological performance tests for the prediction of FRAN (Table 1) showed that blood lactate, HR_max_, HR_av_, and RPE achieved higher values for 2K Row and maximum repetitions of thrusters.

The maximum repetition of thrusters and pull-ups, and 1RM of thrusters resulted in moderate to strong negative correlation with FRAN performance (Figure 2). 2K Row resulted in a moderate positive correlation to FRAN (*r =* 0.628, *p <* 0.05) and a moderate positive correlation to both maximal strength and strength endurance of pull-ups (*r =* 0.531, *p <* 0.05; *r =* 0.697, *p <* 0.05) and thrusters (*r =* 0.532, *p <* 0.05; *r =* 0.576, *p <* 0.05). There were no correlation between RPE, blood lactate, and FRAN performance (Table 2).

## 4. Discussion

CrossFit is one of the most popular training programs but there is a tremendous lack of research that needs to be fulfilled [6,8,9,17,18,19]. Hence, explanations for physiological and morphological predictors require investigation to clarify the physiological indicators of CrossFit sports performance. The main objective of this study was to analyze the physical and physiological variables of CrossFit athletes performing a FRAN WOD to predict the best performance. Our hypothesis was that the neuromuscular system (maximal and strength endurance) was the predictor of performance. Our results confirmed that the performance determinants of FRAN were maximal and strength endurance of thrusters, strength endurance of pull-ups, and the performance of 2K Row.

During FRAN, the athletes achieved high physiological values of blood lactate, HR_max_, and HR_av_. (Table 1). Similar results [20] were found by FRAN by Fernandez-Fernandez et al. [12] with blood lactate of 14.0 mmol L^−1^, HR_av_ 179 bpm, and RPE 8.4. Maté-Muñoz et al. [4], in a similar strength-based WOD, also reported higher results with HR_av_ 171 bpm, HR_max_ 185 bpm, and blood lactate 11.49 mmol L^–1^. These high values reflected the high intensity of the exercises promoted by FRAN and other similar CrossFit WODs [4,8,9,12,21], but they were no key determinants for a better result in this WOD. The performance time of our study was in the 50–60th percentile of Mangine et al. [17], a result that showed that our athletes were of an intermediate level for CrossFit. In the strength endurance test, pull-ups and thrusters, blood lactate, HR_max_, and HR_av_ showed lower results than FRAN. One of the reasons for that could be the unbroken movement criteria of the test that did not allow for higher values of HR and blood lactate. In the 2K Row, both HR and RPE were similar to FRAN, but the blood lactate was higher, maybe a result of a major contribution of the oxidative pathway for the higher duration of the test [21].

There were few studies about CrossFit performance determinants [8,9,19,21] and some stated that muscular strength had a direct relationship with performance in CrossFit [8,18,19,22]. Butcher et al. [8] demonstrated that FRAN was strongly correlated to strength through CrossFit Total, a benchmark that resulted from the sum of the 1RM load of bench press, back squat, and deadlift, and the performance did not result from physiological variables. Our study reported similar conclusions with a strong correlation of strength endurance of thrusters and moderate correlation with maximal strength of thrusters and strength endurance of pull-ups and 2K Row.

The performance of 2K Row test was correlated to maximal and strength endurance of pull-ups and thrusters. This explained why it was a predictor of the FRAN performance, and with VO_2_, which became a predictor based on longer duration (more contribution of the aerobic system) and primarily because it was a non-stop exercise [21]. The various types of exercises performed in CrossFit workouts, for example, 15 deadlifts performed as fast as possible and non-stop, appeared to be anaerobic dependent. However, when performed with strategized breaks, the aerobic capacity appeared to influence the ability to sustain the effort. FRAN, primary because of its short duration, was a high anaerobic based exercise with a strong correlation with anaerobic threshold and with RER above 1 most of the time [8,9,12,21], which increased the importance of the neuromuscular system and could justify why the aerobic capacity was not a predictor of performance for FRAN. A longer duration of the exercise was one of the reasons that other studies reported the aerobic capacity as a determinant of CrossFit performance, e.g., Farrar et al. [23] reported that 12 min of continuous kettlebell swings resulted in 65% of aerobic system contribution and Bellar et al. [9] stated that a 12-min AMRAP showed a stronger association with the VO_2max_ than with the anaerobic peak power.

The muscular strength correlation with FRAN could be explained by a higher relationship between maximal and strength endurance [24] that, according to Soriano’s et al. [25] meta-analysis, moderate loads from 30 to 70% 1RM seemed to provide the optimal load for power production for squat and bench press. Bellar et al. [9] stated that the whole-body strength was fundamental for the performance of FRAN, explaining 42% of the variance, and Martinez-Gomez et al. [22] reported that absolute and relative 1RM full-squad was associated with the performance in CrossFit movements that involved the lower limbs, especially in relative values. In our data, we found 49% and 64% relative 1RM for thrusters and pull-up, respectively, which resulted in a stronger correlation of strength endurance for thrusters and moderate to stronger correlations for maximal strength of thrusters and strength endurance for pull-ups. Rodriguez et al. [6] reported a correlation between the squat absolute strength and FRAN (*r* = 0.528; *p* < 0.05) due to its similarity to the thruster movement. Our results showed that coaches should prioritize these variables to improve the performance of their athletes during training periodization.

The maximal strength of pull-ups did not have a correlation with the performance of FRAN, and neither with the strength endurance of pull-ups due to the specific strength and the high technical complexity of the movements [8,26]. Rodriguez et al. [6], using the same protocol as our maximal strength pull-up test for the strength endurance pull-up test, reported no correlation of the strength endurance of pull-ups with the performance of FRAN due to the different techniques used in both tests.

This study had some limitations: first, the small sample size; second, the experience and fitness level of the sample that was different from beginners and professional CrossFit athletes, could result in different predictors for FRAN performance. Consequently, our findings must be treated with caution as the results may not be applicable to other CrossFit practitioners; and third, although the participants performed with maximal effort on all the tests and maintained their daily routine, the initial lactate level was not measured. Since the results of the blood lactate were only measured after exercise they must be treated with caution.

## 5. Conclusions

FRAN performance is related strongly to neuromuscular variables. The maximal strength and strength endurance of thrusters and the strength endurance of pull-ups appeared as predictors for a better performance in FRAN WOD, regardless of having a good cardiorespiratory performance and higher values of blood lactate, HR_max_ and HR_av_.

## Figures and Tables

**Figure 1 ijerph-18-04070-f001:**
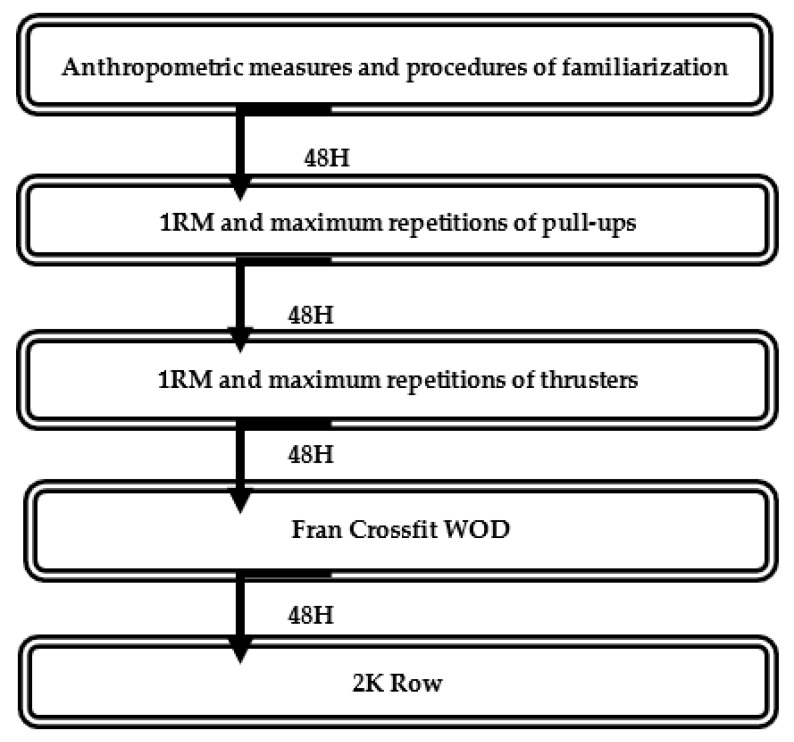
Experimental study design.

**Figure 2 ijerph-18-04070-f002:**
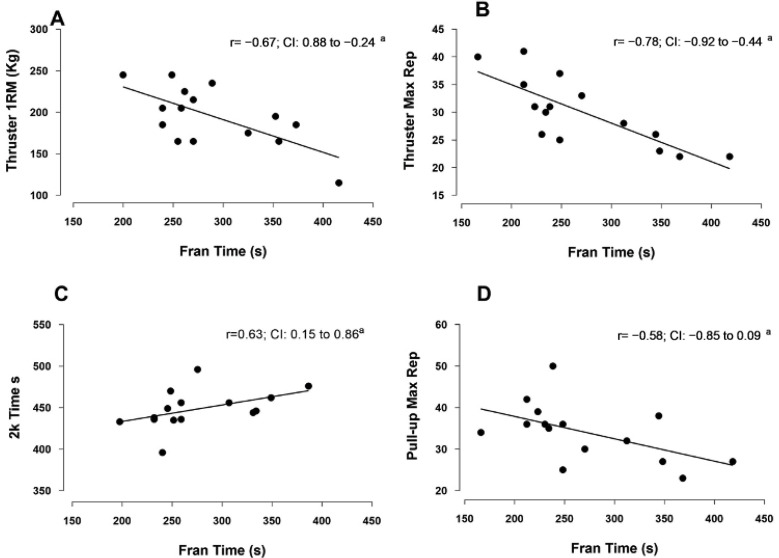
Predictors of FRAN. Correlation between FRAN and (**A**) thruster 1RM (kg); (**B**) thruster maximum repetitions; (**C**) 2K time; and (**D**) pull-up maximum repetitions; ^a^
*p* < 0.05 for correlation.

**Table 1 ijerph-18-04070-t001:** Physiological Performance Tests for the Prediction of FRAN.

Predictors	Median	Interquartile Range
		(25–75)
FRAN(s)	242	217–242
FRAN Blood lactate (mmol L^−1^)	12.6	10.6–15.2
FRAN RPE	10	9–10
FRAN HR_max_ (bpm)	182	179–189
FRAN HR_av_ (bpm)	172	164–183
1RM Pull-up (rep)	123.25	109.99–139.42
1RM Pull-up RPE	7	7–8
1RM Thruster (rep)	88.56	74.93–102.18
1RM Thruster RPE	8	7–8
Maximum Repetitions of Pull-ups (rep)	35	27–38
Maximum Repetitions of Pull-ups RPE	8	8
Maximum Repetitions of Pull-up Blood lactate (mmol L^−1^)	9.7	6.9–11.6
Maximum Repetitions of Pull-ups HR_av_ (bpm)	157.5	151–162
Maximum Repetitions of Pull-ups HR_max_ (bpm)	174	165.25–178.25
Maximum Repetitions Thrusters (rep)	30	25–35
Maximum Repetitions Thrusters RPE	9	8–9
Maximum Repetitions Thrusters Blood Lactate (mmol L^−1^)	10.6	8.9–12.5
Maximum Repetitions Thrusters HR_av_ (bpm)	168.5	162.75–172.75
Maximum Repetitions Thrusters HR_max_ (bpm)	174.5	172–182
2K Row Time(s)	446	436–462
2K Row Blood Lactate (mmol L^−1^)	14.4	10.4–18.1
2K Row RPE	10	9–10
2K Row HR_max_ (bpm)	183	178–186
2K Row HR_av_ (bpm)	168	163–175
2K Row VO_2_ (L/min)	51.96	49.99–53.88

**Table 2 ijerph-18-04070-t002:** Spearman’s correlation of FRAN with Performance Tests for the Prediction of FRAN.

Predictors	FRAN Performance
FRAN Blood Lactate	0.279
FRAN Rate of Perceived Exertion	0.283
FRAN HR_max_	−0.103
FRAN HR_av_	0.130
1RM Pull-up	−0.451
1RM Thruster	−0.608 *
Maximum Repetitions of Pull-ups	−0.598 *
Maximum Repetitions Thrusters	−0.822 *
2K Row Time	0.673 *
2K Row VO_2_	−0.471

* *p* ≤ 0.05.

## Data Availability

The data presented in this study are available on request from the corresponding author.

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
