# Peer review of "Physical and Physiological Predictors of FRAN CrossFit® WOD Athlete’s Performance"

_ijerph, 2021, doi:10.3390/ijerph18084070_

Round 1
Reviewer 1 Report
This study investigated strength and muscular endurance predictors of the CrossFit Fran WOD. I have the following comments and questions below.
Introduction:
- It would be helpful to introduce the Fran WOD in more detail in the intro to provide readers with some context.
- You provide some background about CrossFit and previous studies showing performance predictor variables, which is helpful, but your introduction needs a better rationale for your purpose. Why is this important to study? Please provide more rationale/significance to support the purpose of your study.
- What physiological and psychological variables will you be analyzing?
- Lines 64-65: You mention that you will be assessing the neuromuscular system using maximal strength and maximal endurance. Maximal strength and muscular endurance wouldn’t really be the only indicators of the neuromuscular system, so I wouldn’t use those variables as indicators for the neuromuscular system.
Methods:
- Lines 127-131: You discuss how the pull-up was performed for the test. What about the thrusters?
- Lines 150-151: Please describe how you specifically estimated maximal aerobic capacity. Did you use an equation, and if so, what equation did you use?
- Lines 156-157: What specific type/brand of bioelectrical impedance analysis device did you use?
- You mentioned in your purpose that you would be measuring psychological variables, but I don’t see psychological variables assessed in your methods. What psychological variables are you referring to in your purpose?
Results:
- I don’t see a figure representing 1RM thrusters association with Fran. Was there a significant correlation? If not, I would report that there was not in the results.
- What about blood lactate levels, heart rate measurements, and RPE? What was the purpose of measuring those variables? Did you just use them to compare intensity between the tests and the Fran WOD? Did you look at blood lactate levels in terms of predictors for performance? Please provide more detail in either the methods or results about why those variables were measured and how you used them.
- What about the maximal aerobic capacity that you estimated? You mentioned in the methods that you estimated maximal aerobic capacity, but it is not mentioned in the text or shown in the table or figures in the results.
Discussion:
- So overall, you found that maximal strength and muscular endurance for thrusters and pull-ups were predictors of performance, so why is this important? Please provide more discussion of the significance of your study. And what was the best predictor of performance?
- What about a limitations section? Please provide a section explaining the limitations of the study (i.e. small sample size, fitness level of your participants, etc)
Author Response
We are grateful for your consideration of this manuscript, and we also very much appreciate your suggestions, which have been very helpful in improving the manuscript. We also thank the reviewers for their careful reading of our text. All the comments we received on this study of all reviewers have been attended into account in improving the quality of the article, and we present our reply to each of them separately.
Introduction:
It would be helpful to introduce the Fran WOD in more detail in the intro to provide readers with some context.
R: Thank for the comment. We have added a sentence “Fran WOD is one the most famous and one that every Crossfit athletes does to control their performance improvements. “(LINE:61-63)
You provide some background about CrossFit and previous studies showing performance predictor variables, which is helpful, but your introduction needs a better rationale for your purpose. Why is this important to study? Please provide more rationale/significance to support the purpose of your study.
R:We have provided a better justify for the study.(LINE:61-63)
What physiological and psychological variables will you be analyzing?
R:We have investigated only performance physiological parameters. Although we know that psychological conditions could influence exercise performance, we did not measured these parameters.
Lines 64-65: You mention that you will be assessing the neuromuscular system using maximal strength and maximal endurance. Maximal strength and muscular endurance wouldn’t really be the only indicators of the neuromuscular system, so I wouldn’t use those variables as indicators for the neuromuscular system.
R:Thank for the comment. We used this variables because they are easy and accessible to use by coaches and athletes.
Methods:
Lines 127-131: You discuss how the pull-up was performed for the test. What about the thrusters?
R. We add sentence “the bar started from the floor” (LINE: 145-146)
Lines 150-151: Please describe how you specifically estimated maximal aerobic capacity. Did you use an equation, and if so, what equation did you use?
R. We used the FORMULA. based on mean power (WM) attained in the 2 km test, expressed in watts: VO2max (l/min) in males = 1.682 + 0.0097 WM.
Lines 156-157: What specific type/brand of bioelectrical impedance analysis device did you use?
R. We used BF303 Scale.(LINE:156-159)
You mentioned in your purpose that you would be measuring psychological variables, but I don’t see psychological variables assessed in your methods. What psychological variables are you referring to in your purpose?
R. We purpose to measure physical and physiological variables.
Results:
I don’t see a figure representing 1RM thrusters association with Fran. Was there a significant correlation? If not, I would report that there was not in the results.
R. We add the figure with 1rm Thruster. (LINE:200)
What about blood lactate levels, heart rate measurements, and RPE? What was the purpose of measuring those variables? Did you just use them to compare intensity between the tests and the Fran WOD? Did you look at blood lactate levels in terms of predictors for performance? Please provide more detail in either the methods or results about why those variables were measured and how you used them.
R. We add sentence “There were no correlation between RPE, blood lactate and Fran performance”. (LINE 195-196)
What about the maximal aerobic capacity that you estimated? You mentioned in the methods that you estimated maximal aerobic capacity, but it is not mentioned in the text or shown in the table or figures in the results.
R. We add on the table. (Table 1. LINE 189)
Discussion:
So overall, you found that maximal strength and muscular endurance for thrusters and pull-ups were predictors of performance, so why is this important? Please provide more discussion of the significance of your study. And what was the best predictor of performance?
R. We add sentence explaining that besides the higher values of blood lactate and HR, they aren´t key predictors for a better result in Fran. (LINE 222-223). And sentence about the importance of our results to coaches during periodization (LINE267-268).
What about a limitations section? Please provide a section explaining the limitations of the study (i.e. small sample size, fitness level of your participants, etc)
R. We add “This study has some limitations: first the small sample size; second the experience and fitness level of the sample that is different from beginners and professional Crossfit athletes can result in different predictors for Fran performance; So our findings must be treated with caution as the results may not be applicable to other Crossfit practitioners.”(LINE:275-278)
Reviewer 2 Report
The authors of the present paper have analyzed a relationship between the physical and physiological variables recorded during 1RM & maximum repetitions of pull-ups test, 1RM & maximum repetitions of thrusters as well as 2K test and performance during benchmark workout “Fran”, to identify which variables best determine performance during short benchmark workouts “Fran”, in recreational trained CrossFit athletes. At the end, the authors concluded that “Fran performance is strongly related to neuromuscular variables, maximal and endurance strength training of thrusters”
General criticism of this work.
The authors have mastered analyze the physical and physiological variables of recreational trained CrossFit athletes during one of the most famous WOD, Fran, and they tried to identify which variables best determine performance.
The experimental problem is very interesting. The studies were well planned and carried out but I have major comments and considerations to the statistical method, results and discussion section.
Statistics: For all measures, descriptive statistics were calculated (means and standard deviation). It is not clear why “Spearman correlations were used to analyze all the correlation between study variables”. Means and standard deviation are not appropriate for reporting data analyzed by nonparametric statistics as they will not provide a useful description of the location and range of the data. The structure of the article (statistical methods, results) is difficult to follow.
This makes the reader very uncertain about the results provided.
Fig 2.
The research results, which I believe constitute the essence of this work, are not clearly and carefully presented in Figure 2.
The information presented by the authors in figure 2 is illegible. It seems there is missing information about details in the graph. I could not identify the “pattern" of lines presented in fig 2. There are actually some results in the graph but no information is provided concerning them. There is also no legend notices for this figure.
Perhaps it would be clearer to present the results in the table. In my opinion, the Authors should introduce all spearman correlation coefficients results and take care of the read presentation of the results obtained.
Also, if I got it right: Greater results in absolute time of 2K tests were associated with longer time taken to complete the WOD called "Fran" r=0.63.
On the other hand, greater Maximum repetition of thrusters and pull-ups, and 1RM of thrusters showed associations with less time needed to complete WODs, although the value of "r" was moderate.
Therefore, why the positive (+) or negative (-) correlations were not noticed by the authors and interpreted. The authors should report in their results concerning the direction of studied relationships and I think they should refer to the above-mentioned issues. Mechanisms should be discussed appropriately to establish limitations and domains of applicability of the model and referred to the literature.
An important element of the correct interpretation of the results is the attention to details obtained in their own research and quoting the results of other authors' research without its misinterpretation. I do not understand some of the thoughts include down in the discussion that are not found in connection with the cited literature. 218; 219; 220.
The authors should include a fairly comprehensive limitations section at the end of the discussion.
The conclusion is an over interpretation of the obtained results. The authors have only shown that the total time taken to complete the WOD called "Fran” is strongly related to maximal and endurance strength, maximal strength and strength endurance of thrusters, and strength endurance of pull-ups.
Author Response
We are grateful for your consideration of this manuscript, and we also very much appreciate your suggestions, which have been very helpful in improving the manuscript. We also thank the reviewers for their careful reading of our text. All the comments we received on this study of all reviewers have been attended into account in improving the quality of the article, and we present our reply to each of them separately.
Statistics: For all measures, descriptive statistics were calculated (means and standard deviation). It is not clear why “Spearman correlations were used to analyze all the correlation between study variables”. Means and standard deviation are not appropriate for reporting data analyzed by nonparametric statistics as they will not provide a useful description of the location and range of the data. The structure of the article (statistical methods, results) is difficult to follow.
R.We used Spearman because the data is nonparametric. We add a table with median and interquartile range.(Table 1 LINE:189)
This makes the reader very uncertain about the results provided.
Fig 2. The research results, which I believe constitute the essence of this work, are not clearly and carefully presented in Figure 2.
R.We add x-axis in every graph and legend in figure 2.(LINE:205)
The information presented by the authors in figure 2 is illegible. It seems there is missing information about details in the graph. I could not identify the “pattern" of lines presented in fig 2. There are actually some results in the graph but no information is provided concerning them. There is also no legend notices for this figure. Perhaps it would be clearer to present the results in the table. In my opinion, the Authors should introduce all spearman correlation coefficients results and take care of the read presentation of the results obtained.
R.We add a table with spearman correlation coefficients (table 2 LINE200).
Also, if I got it right: Greater results in absolute time of 2K tests were associated with longer time taken to complete the WOD called "Fran" r=0.63. On the other hand, greater Maximum repetition of thrusters and pull-ups, and 1RM of thrusters showed associations with less time needed to complete WODs, although the value of "r" was moderate.
R. Thank for the comment, we agree.
Therefore, why the positive (+) or negative (-) correlations were not noticed by the authors and interpreted. The authors should report in their results concerning the direction of studied relationships and I think they should refer to the above-mentioned issues. Mechanisms should be discussed appropriately to establish limitations and domains of applicability of the model and referred to the literature.
R. We explained in the text the negative and positive correlations. (LINE 193-197)
An important element of the correct interpretation of the results is the attention to details obtained in their own research and quoting the results of other authors' research without its misinterpretation. I do not understand some of the thoughts include down in the discussion that are not found in connection with the cited literature. 218; 219; 220.The authors should include a fairly comprehensive limitations section at the end of the discussion.
R. We add “This study has some limitations: first the small sample size; second the experience and fitness level of the sample that is different from beginners and professional Crossfit athletes can result in different predictors for Fran performance; So our findings must be treated with caution as the results may not be applicable to other Crossfit practitioners.”(LINE:275-278)
The conclusion is an over interpretation of the obtained results. The authors have only shown that the total time taken to complete the WOD called "Fran” is strongly related to maximal and endurance strength, maximal strength and strength endurance of thrusters, and strength endurance of pull-ups.
R:We add sentence “and higher values of blood lactate, HR máx and HRav.”(LINE 284)
Reviewer 3 Report
Review [IJERPH] Manuscript ID: ijerph-1138428
Title: Physical and physiological predictors of Fran CrossFit 2 ® wod athlete´s performance
It is already generally known that appropriately arranged and selected Crossfit training program is aimed at the development of all major motor skills, i.e. strength, speed, endurance, motor coordination and the most desirable in sports power - also called dynamic or explosive force. The essence of this type of training is properly selected exercises and functional movements that occur in everyday life. The effect of training is an athletic physique, improved well-being and self-confidence. Not surprisingly, this type of training has become extremely popular among different age groups. Therefore, there is a need to find measurable parameters that will help trainers and exercisers not only optimise the training process, but also be able to predict and achieve the desired results.
The authors of this paper took the effort and conducted an interesting experiment to analyse „the physical and physiological variables of recreational trained Crossfit athletes during Fran WOD” and identification of „variables best determine performance. We hypothesized that the neuromuscular system (maximal strength and strength endurance) would be the predictor of performance in FRAN CrossFit ® WOD”.
It should be emphasised that the strongest point of the work is a very reliable description of the experiment. The authors described in detail both the process of preparing the subjects and the course of the test itself, to which the participants were subjected.
There are my comments on other aspects of the work below:
Abstract:
25 line: The authors here use the abbreviation WOD for expression “workout of the day”. However, there is this abbreviation in small letters in the title. Please standardise this.
30 line: The authors write, that „Fifteen Crossfit practitioners performed, on separate days…”. It is not clear whether all the subjects participated simultaneously in the experiment, which lasted for 5 days consecutively, or whether they each did it alone on consecutive days. Please clarify this.
32 line: The expression „Blood lactate” is not accurate. I recommend the use of „blood lactate concentrate or blood level”
33 line: I know that the abbreviation RPE is generally known among researchers in a specific scientific field, but please note that not every reader needs to know it. I recommend an explanation of this abbreviation.
37 line: This is where the expression appears for the first time in the work „neuromuscular variables”. It is not understood which are the parameters investigated in this experiment.
Materials and Method:
71-72 lines: It seems that for the sake of order and precision of the presentation of the anthropometric data of the participants, it would be necessary to give, that „24.03±4.2 years” is “age”, „78.2±10.59 kg” is body weight, „1.75 ±0.07 m” to body height, „25.82±2.7 kg/m2” to BMI a „19.39±4.8 body fat %” is „body fat 19.39±4.8%”. Besides, it is unclear why the authors listed the fat content of the participants' bodies when this parameter was not analysed. In addition, it seems that in this case a much more desirable parameter would be the lean body mass, including muscle mass.
165-166 lines: The authors write that they recorded the results of the analysed parameters only once after a given exercise. It is not understandable why the authors did not record the same parameters before a given exercise in order to analyse the changes and be able to interpret the values obtained for the parameters. This is particularly important when analysing lactic acid concentration. The value recorded only after exercise does not reliably reflect the metabolic response to a given load, especially as the sessions took place at 16.00 – 18.00 (86 line). At this time of day, it was very likely that the subjects had not started a given exercise with physiological levels of lactic acid (< 2mmol/l). It would be very desirable to analyse the change in lactic acid content "before" and "after" exercise. Please explain this issue.
Limitations of paper:
What limitations do the authors of their paper find? Please complete the text of your paper with this section.
Other comments:
256 line: There is „(r = .528; p <.05) and it seems it should be (r = 0.528; p <0.05)
Author Response
We are grateful for your consideration of this manuscript, and we also very much appreciate your suggestions, which have been very helpful in improving the manuscript. We also thank the reviewers for their careful reading of our text. All the comments we received on this study of all reviewers have been attended into account in improving the quality of the article, and we present our reply to each of them separately.
Abstract:
25 line: The authors here use the abbreviation WOD for expression “workout of the day”. However, there is this abbreviation in small letters in the title. Please standardise this. R. We change to wod. (LINE:25-29)
30 line: The authors write, that „Fifteen Crossfit practitioners performed, on separate days…”. It is not clear whether all the subjects participated simultaneously in the experiment, which lasted for 5 days consecutively, or whether they each did it alone on consecutive days. Please clarify this.
R. We add sentence “performed alone” (LINE:30)
32 line: The expression „Blood lactate” is not accurate. I recommend the use of „blood lactate concentrate or blood level”
R. We changed to blood lactate concentrate
33 line: I know that the abbreviation RPE is generally known among researchers in a specific scientific field, but please note that not every reader needs to know it. I recommend an explanation of this abbreviation.
R. The RPE is described in Rating of perceived exertion in the methods.
37 line: This is where the expression appears for the first time in the work „neuromuscular variables”. It is not understood which are the parameters investigated in this experiment.
R. We remove neuromuscular variables and refer only the predictors of performance.
Materials and Method:
71-72 lines: It seems that for the sake of order and precision of the presentation of the anthropometric data of the participants, it would be necessary to give, that „24.03±4.2 years” is “age”, „78.2±10.59 kg” is body weight, „1.75 ±0.07 m” to body height, „25.82±2.7 kg/m2” to BMI a „19.39±4.8 body fat %” is „body fat 19.39±4.8%”. Besides, it is unclear why the authors listed the fat content of the participants' bodies when this parameter was not analysed. In addition, it seems that in this case a much more desirable parameter would be the lean body mass, including muscle mass.
R. We used BF% to characterize the sample.
165-166 lines: The authors write that they recorded the results of the analysed parameters only once after a given exercise. It is not understandable why the authors did not record the same parameters before a given exercise in order to analyse the changes and be able to interpret the values obtained for the parameters. This is particularly important when analysing lactic acid concentration. The value recorded only after exercise does not reliably reflect the metabolic response to a given load, especially as the sessions took place at 16.00 – 18.00 (86 line). At this time of day, it was very likely that the subjects had not started a given exercise with physiological levels of lactic acid (< 2mmol/l). It would be very desirable to analyse the change in lactic acid content "before" and "after" exercise. Please explain this issue.
R.The reason for the measurement of blood lactate after the exercise was to characterize the metabolic response. All the participants performed every exercise without fatigue and maintained their daily routine.
Limitations of paper:
What limitations do the authors of their paper find? Please complete the text of your paper with this section.
R. We add “This study has some limitations: first the small sample size; second the experience and fitness level of the sample that is different from beginners and professional Crossfit athletes can result in different predictors for Fran performance; So our findings must be treated with caution as the results may not be applicable to other Crossfit practitioners.” (LINE: 275-278)
Other comments:
256 line: There is „(r = .528; p <.05) and it seems it should be (r = 0.528; p <0.05).
R. We changed.
Round 2
Reviewer 2 Report
Thank you very much for the response from the authors. I accept almost all the answers and accept the corrections. However, I still think the detailed information is missing. The authors added the results of O2 in Table 1. How and when the authors estimated or measured oxygen uptake [[ VO2 (l/min)] in the 2K Row test? They should make efforts to explain this issue in a manuscript.
Author Response
We are grateful for your consideration of this manuscript, and we also very much appreciate your suggestions, which have been very helpful in improving the manuscript. We also thank the reviewers for their careful reading of our text. All the comments we received on this study of all reviewers have been attended into account in improving the quality of the article, and we present our reply to each of them separately.
Thank you very much for the response from the authors. I accept almost all the answers and accept the corrections. However, I still think the detailed information is missing. The authors added the results of O2 in Table 1. How and when the authors estimated or measured oxygen uptake [[ VO2 (l/min)] in the 2K Row test? They should make efforts to explain this issue in a manuscript.
R: Thank for the comment. We added sentence “Maximum aerobic capacity was estimated according to the work average attained in the performance of 2000m row test in a Concept 2 ergometer [12,13]: VO2 máx(l/min) = 1.631+0.0088Wav” (LINE152-154)
Reviewer 3 Report
Review [IJERPH] Manuscript ID: ijerph-1138428
Title: Physical and physiological predictors of Fran CrossFit 2 ® wod 3 athlete´s performance
Thank you very much for such a comprehensive response from the authors. I accept almost all the answers and accept the corrections.
However, I still think the weakest part of this study is the single data (that is only after exercise) regarding e.g. lactic acid concentration. The authors wrote in response that “The reason for the measurement of blood lactate after the exercise was to characterize the metabolic response. All the participants performed every exercise without fatigue and maintained their daily routine.” With all due respect, having only one measurement - without reference in a situation without stimulus (effort) cannot reliably determine the metabolic response to that stimulus. Also, on what basis did the authors determine that the subjects were not fatigued? I would very much appreciate an explanation of this issue. And if it cannot be explained by adding the results, please put it in the "Limitation" section.
Author Response
We are grateful for your consideration of this manuscript, and we also very much appreciate your suggestions, which have been very helpful in improving the manuscript. We also thank the reviewers for their careful reading of our text. All the comments we received on this study of all reviewers have been attended into account in improving the quality of the article, and we present our reply to each of them separately.
Thank you very much for such a comprehensive response from the authors. I accept almost all the answers and accept the corrections.
However, I still think the weakest part of this study is the single data (that is only after exercise) regarding e.g. lactic acid concentration. The authors wrote in response that “The reason for the measurement of blood lactate after the exercise was to characterize the metabolic response. All the participants performed every exercise without fatigue and maintained their daily routine.” With all due respect, having only one measurement - without reference in a situation without stimulus (effort) cannot reliably determine the metabolic response to that stimulus. Also, on what basis did the authors determine that the subjects were not fatigued? I would very much appreciate an explanation of this issue. And if it cannot be explained by adding the results, please put it in the "Limitation" section.
R: Thank for the comment. We added sentence in the limitation of the study “third, besides the participants performed with maximal effort all the tests and maintained their daily routine the initial lactate level was not measured. So, results of blood lactate only after exercise must be treated with caution.”